# Variable Complexity in the Univariate and Multivariate Structural Causal Model

## Abstract

We provide a critical view of the univariate Structural Causal Model. We show that by comparing the individual complexities of univariate cause and effect, ~~in the Structural Causal Model,~~ one can identify the cause and the effect, without considering their interaction at all. ~~The entropy of each variable is ineffective in measuring the complexity, and We propose~~ We suggest to capture ~~it~~ the complexity by the reconstruction error of an autoencoder that operates on ~~the list of sorted samples~~ the percentiles of the distribution. Comparing the reconstruction errors of the two autoencoders, one for each variable, is shown to perform well on the accepted benchmarks of the field. Hence, the decision as to which of the two is the cause and which is the effect may not be based on causality but on complexity.

In the multivariate case, where one can ensure that the complexities of the cause and effect are balanced, we propose a new adversarial training method that mimics the disentangled structure of the causal model. We extend the results of Zhang & Hyvrinen (2010) to the multidimensional case, showing that such modeling is only likely in the direction of causality. Furthermore, the learned model is shown theoretically to perform the separation to the causal component and to the residual (noise) component. Our multidimensional method obtains a significantly higher accuracy than the literature methods.

## 1 Introduction

A long standing debate in the causality literature, is whether causality can be inferred without intervention (Pearl, 2009; Spirtes et al., 2000). The Structural Causal Model (SCM) (Spirtes et al., 2000) is a simple causative model for which many results demonstrate the possibility of such inference (Stegle et al., 2010; Bloebaum et al., 2018; Goudet et al., 2018; Lopez-Paz et al., 2017; 2015). In this model, the effect ($Y$) is a function of the cause ($X$) and some independent random noise ($E$).

In this work, we take a critical perspective of the univariate SCM. We demonstrate empirically that for the ~~1D~~ univariate case, which is the dominant case in the existing literature, the SCM ~~model~~ leads to an effect that has a lower complexity than the cause. Therefore, one can identify the cause and the effect by measuring their individual complexities, with no need to make the inference based on both variables simultaneously. Thus, the decision as to which of the two is the cause and which is the effect may not be based on causality but on complexity.

Since we are dealing with unordered ~~1D~~ univariate random variables, the complexity measure has to be based on the probability distribution function. As we show empirically, comparing the entropies of the ~~distribution~~ two random variables is ineffective for inferring the causal direction. We, therefore, consider the quantiles, i.e, fixed sized vectors that are obtained as sub-sequences of the sorted sampled values of the variable.

We consider suitable complexity scores for these vectors. In our analysis, we show that the reconstruction error of an autoencoder of a multivariate random variable is a valid complexity measure. In addition, we link the reconstruction error based complexity, in the case of variational autoencoders, to the differential entropy of the input random variable. Hence, by computing the reconstruction errors of trained autoencoders on these vectors we estimate the entropies of the quantile vectors of $X$ and $Y$.

The challenges of measuring causality independently of complexity in the 1D case lead us to consider the multidimensional case, where the complexity can be controlled by, e.g., manipulating the dimension of the noise signal in the SCM ~~model~~. Note that unlike Goudet et al. (2018), we consider pairs of multivariate vectors and not many univariate variables in a graph structure. We demonstrate that for the multidimensional case, any method that is based on comparing the complexity of the individual random variables $X$ and $Y$ fails to infer causality of random variables. Furthermore, we extend the result of Zhang & Hyvrinen (2010) to the multidimensional case and prove that an SCM is unlikely to hold in both directions $X \to Y$ and $Y \to X$ for reasonable conditions.

Based on our observations, we propose a new causality inference method for multidimensional cause and effect. The algorithm learns three networks in a way that mimics the parts of the SCM ~~model~~. The noise part is unknown and is replaced by a function that is constrained to be independent of the cause, as captured by an adversarial loss. However, we show analytically and empirically that even without the explicit constraint, in several cases, such an independence emerges.

Our empirical results support our analysis and demonstrate that in the ~~1D~~ univariate case, assigning cause and effect based on complexity is competitive with the state of the art methods. In the multidimensional case, we show that the proposed method outperforms existing and new extensions of the literature methods to the multidimensional case.

## 1.1 PROBLEM SETUP

We investigate the problem of causal inference from observational data. A non-linear structural causal model (SCM for short) is a generative process of the following form:

$$
\begin{aligned}
X &\sim \mathbb{P}_X \\
E &\sim \mathbb{P}_E \\
Y &\leftarrow g(f(X), E)
\end{aligned}
\tag{1}
$$

The functions $g : \mathbb{R}^{d_f + d_e} \to \mathbb{R}^n$ and $f : \mathbb{R}^n \to \mathbb{R}^{d_f}$ are fixed and unknown. In general, $g$ and $f$ are non-linear. Here, $X$ is the input random variable and $E$ is the environment random variable that is independent of $X$. We say that $X$ causes $Y$ if they satisfy a generative process such as Eq. 1.

We present methods for inferring whether $X$ causes $Y$ (denoted by $X \to Y$) or $Y$ causes $X$, or neither. The algorithm is provided with i.i.d samples $\{(x_i, y_i)\}_{i=1}^m \sim \mathbb{P}_{X,Y}^m$ (the distribution of $m$ i.i.d samples from the joint distribution $\mathbb{P}_{X,Y}$) from the generative process of Eq. 1. In general, by (cf. Prop 4.8, (Peters et al., 2017)), for any joint distribution $\mathbb{P}_{X,Y}$ of two random variables $X$ and $Y$, there is an SCM, $Y = g(f(X), E)$, where $E$ is a noise variable, such that, $X \perp\!\!\!\perp E$ and $h$ is some (measurable) function. Therefore, in general, deciding whether $X$ causes $Y$ or vice versa is ill-posed when only provided with samples from the joint distribution. However, Zhang & Hyvrinen (2010) showed for the one dimensional case (i.e., $X, Y \in \mathbb{R}$) that under reasonable conditions, a representation $Y = g(f(X) + E)$ holds only in one direction. In Sec. 3.2 we extend this theorem and show that a representation $Y = g(f(X), E)$ holds only in one direction when $g$ and $f$ are assumed to be neural networks and $X, Y$ are multidimensional (we call such SCMs neural SCMs).

Throughout the paper, we denote by $\mathbb{P}_U[u] := \mathbb{P}[U \leq u]$ the cumulative distribution function of a uni/multi-variate real valued random variable $U$ and $\mathbb{P}$ is a standard Lebesgue measure. Additionally, we denote by $p_U(u) = \frac{d}{du} \mathbb{P}_U[u]$ the probability density function of $U$ (if exists, i.e., $\mathbb{P}_U[u]$ is absolutely continuous). We denote by $\mathbb{E}_{u \sim U}[f(u)]$ or by $\mathbb{E}_{p_U}[f(u)]$ the expected value of $f(u)$ for $u$ that is distributed by $\mathbb{P}_U[u]$, depending on the context. The identity matrix of dimension $n \times n$ is denoted by $I_n$. We recall a few notations from information theory (Shannon, 1948; Cover & Thomas, 2006). The entropy of a random variable $U$, is defined as $h(U) := -\mathbb{E}_{p_U}[\log(p_U(u))]$, where $\mathcal{U}$ is the domain of $U$. The mutual information between two random variables $U$ and $V$ is denoted by $I(U; V) := \mathbb{E}_{p_{(U,V)}}\left[\log\left(\frac{p_{U,V}(u,v)}{p_U(u)p_V(v)}\right)\right]$. Sometimes, we will denote $I_p(U; V)$, to specify the PDF, $p$, of the variables $(U, V)$.

## 1.2 RELATED WORK

In causal inference, the algorithm is provided with a dataset of matched samples $(x, y)$ of two random variables $X$ and $Y$ and decides whether $X$ causes $Y$ or vice versa. The early wisdom

in this area asserted that this asymmetry of the data generating process (i.e., that $Y$ is computed from $X$ and not vice versa) is not apparent from looking at $\mathbb{P}_{X,Y}$ alone. That is, in general, provided with samples from the joint distribution $\mathbb{P}_{X,Y}$ of two variables $X, Y$ does tell us whether it has been induced by an SCM from $X$ to $Y$ or from $Y$ to $X$.

In publications such as (Pearl, 2009; Spirtes et al., 2000) it is argued that in order to decide whether $X$ causes $Y$ or vice versa, one needs to observe the influence of interventions on the environment parameter. To avoid employing interventions, most publications assume prior knowledge on the generating process and/or independence between the cause and the mechanism.

Various methods for causal inference under the SCM ~~model~~ have been suggested. Many of these methods are based on independence testing, where the algorithm models the data as $Y = g(f(X), E)$ (and vice versa) and decides upon the side that provides a better fitting in terms of mapping accuracy and independence between $f(X)$ and $E = r(X, Y)$. The LiNGAM (Shimizu et al., 2006) algorithm assumes that the SCM takes the form $Y = \beta X + E$, where $X \perp\!\!\!\perp E$ and $\beta \in \mathbb{R}$ and $E$ is non-Gaussian. The algorithm learns $\beta$, such that, $X$ and $Y - \beta X$ are independent by applying independent component analysis (ICA for short). The Direct-LiNGAM (Shimizu et al., 2011) extends this method and replaces the mutual information minimization with a non-parametric kernel based loss (Bach & Jordan, 2003). However, the computation of this loss is of order $\Theta(m^2)$ in the the worst case ($m$ is the number of samples).

The ANM approach (Hoyer et al., 2009) extends LiNGAM's modeling and assumes that $Y = f(X) + E$, where $X \perp\!\!\!\perp E$. In their modeling they employ a Gaussian process for the learned mechanism between the two random variable. The function $f$ is trained to map between $X$ and $Y$ (and vice versa) and the method then tests whether, $X$ and $f(X) - Y$ are independent. The independence test is based on kernels (Gretton et al., 2005).

A different extension of LiNGAM is the PNL algorithm by Zhang & Hyvrinen (2010). This algorithm learns a mapping between $X$ and $Y$ (and vice versa) of the form $Y = g(f(X) + E)$, where $f(X)$ and $E$ are restricted to be independent. To do so, PNL trains two neural networks $f$ and $g$ to minimize the mutual information between $f(X)$ and $E = g^{-1}(Y) - f(X)$. The main disadvantage of this methods is the reliance on the minimization of the mutual information. It is often hard to measure and optimize the mutual information directly, especially in higher dimensions. In many cases, it requires having an explicit modeling of the density functions, because of the computation of expected log-probability within the formulation of the entropy measure.

In our method, we take a similar approach to the above methods. However, our independence constraint is non-parametric and is applied on the observations rather on an explicit modeling of the density functions. In addition, we do not assume restrictive structural assumptions and treat the generic case where the effect is of the form $Y = g(f(X), E)$. In addition, our method is computationally efficient.

Several other methods take different approaches. For example, the Information Geometric Causal Inference algorithm (IGCI) (Daniusis et al., 2012) approach determines the causal relationship in a deterministic setting $Y = f(X)$ under an independence assumption between the cause $X$ and the mechanism $f$, $\mathrm{Cov}(\log f'(x), p_X) = 0$.

The Conditional Distribution Similarity Statistic (CDS) (Fonollosa, 2016) measures the standard deviation of the values of $Y$ (resp. $X$) after binning in the $X$ (resp. $Y$) direction. The lower the standard deivation, the more likely the pair to be $X \rightarrow Y$. The CURE algorithm (Sgouritsa et al., 2015) compares between $X \rightarrow Y$ and $Y \rightarrow X$ directions in the following manner: is given as follows: if we can estimate $p_{X|Y}$ based on samples from $p_Y$ more accurately than $p_{Y|X}$ based on samples from $p_X$, then $X \rightarrow Y$ is inferred.

The BivariateFit method learns a Gaussian process regressor in both directions and decides upon the side that had the lowest error. The RECI method (Bloebaum et al., 2018) trains a regression model in both directions, and returns the side that produced a lower MSE loss. The utilized regression models were: a logistic function, polynomial functions, support vector regression and neural networks. The CGNN algorithm (Goudet et al., 2018) uses the Maximum Mean Discrepancy (MMD) distance between the distribution produced by modeling $Y$ as an effect of $X$, $(X, g(X, E))$ (and vice versa), and the ground truth distribution. The algorithm compares the two distances and returns the direction that led to a smaller distance. The Gaussian Process Inference model (GPI) (Stegle et al., 2010)

builds two generative models, one for $X \to Y$ and one for $Y \to X$. The distribution of the candidate cause variable is modelled as a Gaussian mixture model, and the mechanism $f$ is a Gaussian process. The causal direction is determined from the generative model that best fits the data.

Finally, it is worth mentioning that several other methods, such as (Heinze-Deml et al., 2017; Zhang et al., 2011) assume a different type of SCM, where the algorithm is provided with separate datasets that correspond to different environments, i.e., sampled i.i.d from $\mathbb{P}_{X,Y|E}$, where the value of $E$ is fixed for all samples in the dataset. In these publications, a different independence condition is assumed: $Y$ is independent of $E$ given $X$. This assumption typically fails in our setting. In our paper, we focus on a vanilla SCM, where the algorithm is only provided with observational samples of $X$ and $Y = g(f(X), E)$ (i.i.d samples). The samples are not divided into subsets that are invariant w.r.t $E$.

## 2 THE UNIVARIATE CASE

In this section, we intend to show that the univariate case is a bit simplistic. For this purpose, we start by describing a method for identifying the cause and the effect, which considers each of the two variables independently without considering the mapping between them. The method is based on the statistics of the quantiles of each of the two distributions. In Sec. 2.2, we show that the reconstruction error of an autoencoder is directly linked to the ~~differential~~ entropy of the distribution it models.

### 2.1 A METHOD FOR INFERRING DIRECTIONALITY

~~We hypothesize that due to the information processing inequality, the entropy of the cause $h(X)$ in the SCM model is higher than the entropy $h(Y)$ of the effect. However, estimating the entropy of each random variable from its samples does not present a consistent difference between the entropies $h(X)$ and $h(Y)$.~~

~~Our method, therefore,~~ The proposed method computes ~~an alternative~~ a complexity score for $X$ and, independently, for $Y$. It then compares the scores and decides that the cause is the random variable that has the larger score among them.

Our scoring function has a few stages. As a first step, it converts the random variable at hand (say, $X$) into a multivariate random variable. This is done by sorting the samples of the random variable, and then cutting the obtained list into fixed sized vectors of length $k$. We discard the largest measurements in the case where the number of samples is not a multiple of $k$. We denote the random variable obtained this way by $U$.

At the second stage, the method trains an autoencoder $A : \mathbb{R}^k \to \mathbb{R}^k$ on these quantile vectors. Formally, $A$ is trained to minimize the following objective:

$$\mathcal{L}_{\text{recon}}(A) := \mathbb{E}_{u \sim U}[\ell(A(u), u)] \tag{2}$$

where $\ell(a, b)$ is some loss function. In our implementation, we employ the $L_2$-loss function, defined as $\ell(a, b) = \|a - b\|_2^2$. Finally, the method uses the value of $\mathcal{L}_{\text{recon}}(A)$, which we refer to as the AEQ score, as a proxy for the complexity of $X$ (smaller loss means lower complexity). It decides that $X$ or $Y$ is the cause, based on which side provides a higher AEQ.

### 2.2 RECONSTRUCTION ERROR AS AN ESTIMATOR OF THE ENTROPY

Next, we would like to provide better understanding of the AEQ method. Since we use the reconstruction error as a measure of complexity, we next link it to entropy, which is an acceptable complexity measure. This will explain that the AEQ method estimates and compares the entropies of the percentiles of $X$ and $Y$. This does not imply that the AEQ method compares between the entropies of $X$ and $Y$.

The analysis ~~This~~ is done in the context of a learned variational autoencoder (VAE) (Kingma & Welling, 2014), with a fixed latent space dimension.

We are given a real-valued vector-random-variable $U$ (not necessarily quantiles) with a PDF $p_U(u)$. We recall the setting of VAEs and the analysis of Zhao et al. (2017). The VAE model employs

a $d$ dimensional latent variable $z \sim \mathcal{N}(0, I_d)$, and considers the family of the distribution $p_\theta(u|z)$ parameterized by $\theta \in \Theta$. The objective of this framework is to select $\theta$ that maximizes the likelihood of the samples of $U$ within $p_\theta(u|z) \cdot p(z)$,

$$\mathbb{E}_{u \sim U} \left[ \log p_\theta(u) \right] = \mathbb{E}_{u \sim U} \left[ \int_{\mathbb{R}^d} p_\theta(u|z) \cdot p(z) \, \mathrm{d}z \right] \tag{3}$$

where $p(z) = \exp(-\|z\|_2^2/2) \cdot (2\pi)^{-k/2}$ is the PDF of $z$ and $p_\theta(u) = p_\theta(u|z) \cdot p(z)$. In typical situations, we select $p_\theta$ to be the PDF of a normal distribution $\mathcal{N}(f_\theta(z), \sigma_1^2 \cdot I_n)$, where $f_\theta(z) \in \mathcal{F}$ is a learned neural network and $\sigma_1^2$ is a variance hyperparameter of the framework. We denote by $\mathcal{P} = \{p_\theta(u|z)|z \in \mathbb{R}^d, \ \theta \in \Theta\}$ the class of density functions over $\mathcal{X}$.

To optimize this objective, it is required to evaluate $p_\theta^{-1}(u)$, which involves the computation of a high dimensional integral. Even though this integral can be approximated by sampling, this is a costy process that has to be done for every sample $u$. To solve this problem, we have a second distribution that is responsible to model the latent codes $z$ that are most probable given $u$. This PDF is denoted by $q_\phi(z|u)$ and corresponds to a normal distribution $\mathcal{N}(g_\phi(u), \sigma_2^2 \cdot I_d)$. Here, $g_\phi(u)$ is a neural network with parameters $\phi \in \Phi$.

Suppose we are given a fixed inference distribution $q_\phi(z|u)$, which maps (probabilistically) inputs $u$ to features $z$. We denote by $q_\phi(u, z) := p_U(u) \cdot q_\phi(z|u)$ the joint distribution of $u$ and $z$, the marginal $q_\phi(z) := \int_u q_\phi(u, z) \mathrm{d}u$ and the posterior $q_\phi(u|z) = q_\phi(z|u) \cdot p_U(u)/q_\phi(z)$.

The optimization criterion is redefined as follows, where for each $z$ we use a member of a class of distributions, $\mathcal{P}$, to fit a different $q_\phi(u|z)$ rather than the entire $p_U$.

$$\mathcal{L} := \mathbb{E}_{q_\phi(u,z)} \left[ \log p_\theta(u|z) \right] \tag{4}$$

The following lemma links the reconstruction loss and the entropy (all proofs are in the appendix)

**Lemma 1.** *In the setting of Sec. 2.2. Let $\theta^*$ be the global optimum of Eq. 4 for a sufficiently large $\mathcal{F}$. Assume that $\mathcal{P}$ is sufficiently large, such that:*

$$\forall z \in \mathcal{Z}, \phi \in \Phi : q_\phi(u|z) \in \mathcal{P} \tag{5}$$

*Then,*
$$h(U) = \frac{1}{2\sigma_1^2} \cdot \mathbb{E}_{u \sim U} \left[ \mathbb{E}_{\epsilon \sim \mathcal{N}(0, I_d)} \left[ \|f_{\theta^*}(g_\phi(u) + \sigma_2^2 \cdot I_d \cdot \epsilon) - u\|_2^2 \right] \right]$$
$$+ \frac{1}{2} \mathbb{E}_{u \sim U} \left[ \|g_\phi(u)\|_2^2 \right] + \frac{d \cdot (\sigma_2^2 - 1 - \log(\sigma_2^2))}{2} + \frac{n}{2} \log(2\pi\sigma_1^2) \tag{6}$$

Therefore, from this lemma, we observe that if $g_\phi$ is learned, such that, the expected norm of the encodings of $u$, $\mathbb{E}_{u \sim U} \left[ \|g_\phi(u)\|_2^2 \right]$ is bounded, then, the differential entropy $h(U)$ is proportional reconstruction error of the learned autoencoder. In particular, for two random variables $U_1$ and $U_2$, their entropies can be compared, by considering the reconstruction error of their autoencoders with the same hyperparameters (latent space dimension $d$, $\sigma_1^2$, $\sigma_2^2$, etc').

## 3  THE MULTIVARIATE CASE

For the univariate case, one can consider the complexity of the $X$ and $Y$ variables of the SCM ~~model~~ and infer directionality. We propose the AEQ complexity for this case, since more conventional complexities are ill-defined for unordered 1D data or, in the case of entropy, found to be ineffective.

In the multivariate case, one can consider the complexity of the random variables in various ways. We focus on the family of complexity measures $C(X)$ that satisfy the assumption that when $X$ and $Y$ are independent

$$C(X, Y) \geq \max(C(X), C(Y)). \tag{7}$$

Sample complexity measures that satisfy this condition are the Shannon Entropy and the Kolmogorov Complexity. Lem. 3 in Appendix. B.2 shows that the complexity that is based on autoencoder modeling is also in this family.

The next result shows that for any complexity measure $C$ in this family, one cannot infer directionality in the multivariate SCM based on $C$.

**Lemma 2.** *Let $C$ be a complexity measure of multivariate random variables (i.e, non-negative and satisfies Eq. 7). Then, there are triplets of random variables $(X, E, Y)$ and $(\hat{X}, E, Y)$, such that, $Y = g(X, E)$, $Y = g'(\hat{X}, E)$, $C(X) < C(Y)$ and $C(\hat{X}) > C(Y)$. Therefore, $C$ cannot serve as a score for causal inference.*

### 3.1 A METHOD FOR CAUSAL INFERENCE

The causality inference algorithm trains neural networks $G, F, R$ and $D$. The success of fitting these networks serves as the score for the causality test.

The function $F$ models the function $f$, $G$ models $g$ and $R(Y)$ aims to model the environment parameter $E$. In general, our method aims at solving the following objective:

$$\min_{G,F,R} \mathcal{L}_{\text{err}}(G, F, R) := \frac{1}{m} \sum_{i=1}^{m} \|G(F(a_i), R(b_i)) - b_i\|_2^2 \tag{8}$$

$$\text{s.t: } A \perp\!\!\!\perp R(B)$$

where $A$ is either $X$ or $Y$ and $B$ is the other option, and $a_i = x_i, b_i = y_i$ or $a_i = y_i, b_i = x_i$ accordingly. To decide whether $X \to Y$ or vice versa, we train a different triplet $G, F, R$ for each direction and see if we can minimize the translation error subject to independence. We decide upon a specified direction if the loss can be minimized subject to independence. In general, searching within the space of functions that satisfy $A \perp\!\!\!\perp R(B)$ is an intractable problem. However, we can replace it with a loss term that is minimized when $A \perp\!\!\!\perp R(B)$.

**Independence loss**  We would like $R(B)$ to capture the information encoded in $E$. Therefore, restrict $R(B)$ and $A$ to be independent in each other. We propose an adversarial loss for this purpose, which is a modified version of a loss proposed by Brakel & Bengio (2017) and later analyzed by (Press et al., 2019).

This loss measures the discrepancy between the joint distribution $\mathbb{P}_{A,R(B)}$ and the product of the marginal distributions $\mathbb{P}_A \times \mathbb{P}_{R(B)}$. Let $d_F$ ($d_R$) be the dimension of $F$'s output ($R$). To measure the discrepancy, we make use of a discriminator $D : \mathbb{R}^{n+d_R} \to [0, 1]$ from a class $\mathcal{C}$ that maximizes the following term:

$$\mathcal{L}_D(D; R) := \frac{1}{m} \sum_{i=1}^{m} \ell(D(a_i, R(b_i)), 1) + \frac{1}{m} \sum_{i=1}^{m} \ell(D(\hat{a}_i, R(\hat{b}_i)), 0) \tag{9}$$

where $D$ is a discriminator network, and $l(p, q) = -(q \log(p) + (1 - q) \log(1 - p))$ is the binary cross entropy loss for $p \in [0, 1]$ and $q \in \{0, 1\}$. In addition, $\{(\hat{a}_i, \hat{b}_i)\}_{i=1}^{m}$ are i.i.d samples from $\mathbb{P}_A \times \mathbb{P}_B$. To create such samples, we can simply generate samples from $\hat{a}_i \sim \mathbb{P}_A$ and $\hat{b}_i \sim \mathbb{P}_B$ independently and then arbitrarily match them into couples $(\hat{a}_i, \hat{b}_i)$.

To restrict that $R(B)$ and $A$ are independent, we train $R$ to fool the discriminator $D$ and to convince it that the two sets of samples are from the same distribution,

$$\mathcal{L}_{\text{indep}}(R; D) := \frac{1}{m} \sum_{i=1}^{m} \ell(D(a_i, R(b_i)), 1) + \frac{1}{m} \sum_{i=1}^{m} \ell(D(\hat{a}_i, R(\hat{b}_i)), 1) \tag{10}$$

**Full objective**  The full objective of our method is then translated into the following program:

$$\min_{G,F,R} \mathcal{L}_{\text{err}}(G, F, R) + \lambda_1 \cdot \mathcal{L}_{\text{indep}}(R; D)$$
$$\min_{D} \mathcal{L}_D(D; R) \tag{11}$$

Where $\lambda_1$ is some positive constant. The discriminator $D$ minimizes the loss $\mathcal{L}_D(D; R)$ concurrently with the other networks. We denote by $c_{\text{real}}$ the percentage of samples $(a_i, b_i)$ that the discriminator classifies as 1 and by $c_{\text{fake}}$ the percentage of samples $(\hat{a}_i, \hat{b}_i)$ that are classified as 0. We note that when $c_{\text{real}} + c_{\text{fake}} \approx 1$, the discriminator is unable to discriminate between the two distributions, i.e., it is wrong in classifying half of the samples.

Our method decides if $X$ causes $Y$ or vice versa by comparing the score $\mathcal{L}_{\text{err}}(G, F, R)$. A lower error means a better fit. The full description of the architecture employed for the encoders, generator and discriminator is given in Appendix. A.

## 3.2 ANALYSIS

In this section, we analyze the proposed method. In Thm. 1, we show that if $X$ and $Y$ admit a SCM in one direction, then it admits a SCM in the opposite direction only if the involved functions satisfy a specific partial differential equation.

**Theorem 1** (Identifiability of neural SCMs). *Let $\mathbb{P}_{X,Y}$ admit a neural SCM from $X$ to $Y$ as in Eq. 1, such that $p_X$, and the activation functions of $f$ and $g$ are three-times differentiable. Then it admits a neural SCM from $Y$ to $X$ only if $p_X$, $f$, $g$ satisfy Eq. 34 in the Appendix.*

This result generalizes the one-dimensional case presented in (Zhang & Hyvrinen, 2010), where a one-dimensional version of this differential equation is shown to hold in the analog case.

In the following theorem, we show that minimizing the proposed losses is sufficient to recover the different components, i.e., $F(X) \propto f(X)$ and $R(Y) \propto E$, where $A \propto B$ means that $A = f(B)$ for some invertible function $f$.

**Theorem 2** (Uniqueness of Representation). *Let $\mathbb{P}_{X,Y}$ admit a nonlinear model from $X$ to $Y$ as in Eq. 1, i.e., $Y = g(f(X), E)$ for some random variable $E \perp\!\!\!\perp X$. Assume that $f$ and $g$ are invertible. Let $G$, $F$ and $R$ be functions, such that, $\mathcal{L}_{\mathrm{err}} := \mathbb{E}_{(x,y)\sim(X,Y)}[\|G(F(x), R(y)) - y\|_2^2] = 0$ and $G$ and $F$ are invertible functions and $X \perp\!\!\!\perp R(Y)$. Then, $F(X) \propto f(X)$ and $R(Y) \propto E$.*

Here, $\mathcal{L}_{\mathrm{err}} := \mathbb{E}_{(x,y)\sim(X,Y)}[\|G(F(x), R(y)) - y\|_2^2]$ is the mapping error proposed in Eq. 8. In addition, the assumption $X \perp\!\!\!\perp R(Y)$ is sufficed by the independence loss. In Lem. 6 in Appendix. B.3 we extend this lemma to the case where the mapping loss is not necessarily zero and get rid of the assumption that $G$ is invertible. In this case, instead of showing that $R(Y) \propto E$, we provide an upper bound on the reconstruction of $E$ out of $R(Y)$ (and vice versa) that improves as $\mathcal{L}_{\mathrm{err}}$ decreases.

We note that when the entropy of $R(Y)$ is smaller than the entropy of $E$, then, the independence $X \perp\!\!\!\perp R(Y)$ follows, even without the discriminator. Typically, when $R$ has a limited capacity (i.e., small number of layers or output dimension), then, the entropy of $R(Y)$ is limited as well.

**Theorem 3** (Emergence of independent representations). *Let $\mathbb{P}_{X,Y}$ admits a nonlinear model from $X$ to $Y$ as in Eq. 1, i.e., $Y = g(f(X), E)$ for some random variable $E \perp\!\!\!\perp X$. Assume that $X$ and $Y$ are discrete random variables. Let $G$, $F$ and $R$ be functions, such that, $\mathcal{L}_{\mathrm{err}} := \mathbb{E}_{(x,y)\sim(X,Y)}[\|G(F(x), R(y)) - y\|_2^2] = 0$ and $G$ is an invertible function. Assume that $h(R(Y)) \leq h(E)$ and that $g$ is invertible. Then, we have: $F(X) \perp\!\!\!\perp R(Y)$, $F(X) \propto f(X)$ and $R(Y) \propto E$.*

To conclude this section, by Thm. 1, under reasonable assumptions, if $X$ and $Y$ admit a multivariate SCM in direction $X \to Y$, then, there is no such representation in the other direction. By Thm. 2, by training our method in both directions, one is able to capture the causal model in the correct direction. This is something that is impossible to do in the other direction by Thm. 1. By Thm. 3, we show that in cases where $h(R(Y)) \leq h(E)$, the independence between $F(X)$ and $R(Y)$ emerges implicitly. This is the typical case, when $R$'s capacity is limited.

## 4 EXPERIMENTS

This section is divided into two parts. The first, is devoted to showing that causal inference in the one-dimensional case highly depends on the complexities of the distributions of $X$ and $Y$. In the second part of this section, we show that our multivariate causal inference method outperforms existing baselines. Most of the baseline implementations were taken from the Causality Discovery Toolbox of Kalainathan & Goudet (2019). The experiments with PNL (Zhang & Hyvrinen, 2010), LiNGAM (Shimizu et al., 2006) and GPI (Stegle et al., 2010) are based on their original matlab code.

### 4.1 ONE-DIMENSIONAL DATA

We compared the autoencoder method on several well-known one dimensional cause-effect pairs datasets. Each dataset consists of a list of pairs of real valued random variables $(X, Y)$ with their direction 1 or 0 depending on $X \to Y$ or $Y \to X$ (resp.). For each pair, we have a dataset of samples $\{(x_i, y_i)\}_{i=1}^m$.

Five cause-effect inference datasets, covering a wide range of associations, are used. CE-Net (Goudet et al., 2018) contains 300 artificial cause-effect pairs generated using random distributions as causes, and neural networks as causal mechanisms. CE-Gauss contains 300 artificial cause-effect pairs as generated by Mooij et al. (2016), using random mixtures of Gaussians as causes, and Gaussian process priors as causal mechanisms. CE-Multi (Goudet et al., 2018) contains 300 artificial cause-effect pairs built with random linear and polynomial causal mechanisms. In this dataset, simulated additive or multiplicative noise is applied before or after the causal mechanism.

The real-world datasets include the diabetes dataset by Frank & Asuncion (2010), where causality is from Insulin $\rightarrow$ Glucose. Glucose curves and Insulin dose were analysed for 69 patients, each serves as a separate dataset. To match the literature protocols, the pairs are taken in an orderless manner, ignoring the time series aspect of the problem. Finally, the Tübingen cause-effect pairs dataset by Mooij et al. (2016) is employed. This dataset is a collection of 100 heterogeneous, hand-collected, real-world cause-effect samples.

The autoencoder employed in our method is a fully-connected 5-layered neural network with 3 layers for the encoder and 2 layers for the decoder. The hyperparameters of this algorithm are the sizes of each layer, the activation function and the input dimension, i.e., length of sorted cuts (denoted by $k$ in Sec. 2). Throughout the experiments, we noticed that the hyperparameter that has the highest influence is the input dimension. For all datasets results are stable in the range of $200 \leq k \leq 300$, and we therefore use $k = 250$ throughout the experiments. For all dataset we employed the ReLU activation function, except the Tübingen dataset, where the sigmoid activation function produced better results (results are also reasonable with ReLU, but not state of the art).

In addition to our method, we also employ the entropy of each individual variable as a complexity method. This is done by binning the values of the variables into 50 bins. Other numbers of bins produce similar results.

Tab. 1 presents the mean AUC for each literature benchmark. As can be seen, the AEQ complexity measure produces reasonable results in comparison to the state of the art methods, indicating that the 1D SCM ~~model~~ can be overcome by comparing per-variable scores. On the popular Tübingen dataset, the ~~one-sided~~ AEQ computation outperforms all literature methods.

Tab. 2 presents accuracy rates for various methods on the Tübingen dataset, where such results are often reported in the literature. As can be seen, our simple method outperforms almost all other methods, including methods that employ supervised learning of the cause-effect relation.

## 4.2 MULTIVARIATE DATA

We compared our method on several synthetic datasets. Each dataset consists of a list of pairs of real multivariate random variables $(X, Y)$ with their direction 1 or 0, depending on $X \rightarrow Y$ or $Y \rightarrow X$ (resp.). For each pair, we have a dataset of samples $\{(x_i, y_i)\}_{i=1}^m$.

We employ five datasets, covering multiple associations. Each dataset contains 300 artificial cause-effect pairs. The cause random variable is of the form $X = h(z)$, where $h$ is some function and $z \sim \mathcal{N}(0, \sigma_1^2 \cdot I_n)$. The effect is of the form $Y = g(u(X, E))$, where $E \sim \mathcal{N}(0, \sigma_2^2 \cdot I_n)$ is independent of $X$, $u$ is a fixed function that combined the cause $X$ and the noise term $E$ and $g$ is the causal mechanism. For each dataset, the functions $h$ and $g$ are taken from the same family of causal mechanisms $\mathcal{H}$. Each pair of random variables is specified by randomly selected functions $h$ and $g$.

MCE-Poly is generated element-wise polynomials composed on linear transformations as mechanisms and $u(X, E) = X + E$. MCE-Net pairs are generated using neural networks as causal mechanisms and $u$ is the concatenation operator. The mechanism in MCE-SigMix consists of linear transformation followed by element wise application of $q_{a,b,c}(x) := ab(\tilde{x} + c)/1 + |b \cdot (\tilde{x} + c)|$, where $a, b, c$ are random real valued numbers, which are sampled for each pair and $\tilde{x} = x + e$, where $e$ is the environment random variable. In this case, $u(X, E) = X + E$.

In order to create each one of the synthetic datasets we took the standard synthetic data generators of Kalainathan & Goudet (2019) and extended them to produce multivariate causal pairs. We noticed that a-priori, the produced datasets are imbalanced in a way that the reconstruction error of a standard autoencoder on each random variable can be employed as a score that predicts the cause variable with a high accuracy. Therefore, in order to create balanced datasets, we varied the amount of noise

Table 1: Mean AUC rates of various baselines on different one dimensional cause-effect pairs datasets. Our simple AEQ algorithm achieves competitive results on most datasets.

| Method | Dataset | | | | |
|---|---|---|---|---|---|
| | CE-Net | CE-Gauss | CE-Multi | Tübingen | Diabetes |
| BivariateFit | 77.6 | 36.3 | 55.4 | 58.4 | 0.0 |
| LiNGAM (Shimizu et al., 2006) | 43.7 | 66.5 | 59.3 | 39.7 | **100.0** |
| CDS (Fonollosa, 2016) | 89.5 | 84.3 | 37.2 | 59.8 | 12.0 |
| IGCI (Daniusis et al., 2012) | 57.2 | 33.2 | 80.7 | 62.2 | **100.0** |
| ANM (Hoyer et al., 2009) | 85.1 | 88.9 | 35.5 | 53.7 | 22.2 |
| PNL (Zhang & Hyvrinen, 2010) | 75.5 | 83.0 | 49.0 | 68.1 | 28.1 |
| GPI (Stegle et al., 2010) | 88.4 | **89.1** | 65.8 | 66.4 | 92.9 |
| RECI (Bloebaum et al., 2018) | 60.0 | 64.2 | 85.3 | 62.6 | 95.4 |
| CGNN (Goudet et al., 2018) | **89.6** | 82.9 | **96.6** | 79.8 | 34.1 |
| Entropy as a complexity measure | 49.6 | 49.7 | 50.8 | 54.5 | 53.4 |
| Our AEQ comparison | 62.5 | 71.0 | 96.0 | **82.8** | 95.0 |

Table 2: Accuracy rates of various baselines on the CE-Tüb dataset. Our simple algorithm AEQ achieves almost SOTA accuracy.

| Method | Supervised | Acc |
|---|---|---|
| LiNGAM (Shimizu et al., 2006) | - | 44.3% |
| BivariateFit | - | 44.9% |
| Entropy as a complexity measure | - | 52.5% |
| IGCI (Daniusis et al., 2012) | - | 62.6% |
| CDS (Fonollosa, 2016) | - | 65.5% |
| ANM (Hoyer et al., 2009) | - | 59.5% |
| CURE (Sgouritsa et al., 2015) | - | 60.0%[a] |
| GPI (Stegle et al., 2010) | - | 62.6% |
| PNL (Zhang & Hyvrinen, 2010) | - | 66.2% |
| CGNN (Goudet et al., 2018) | - | 74.4% |
| RECI (Bloebaum et al., 2018) | - | 77.5% |
| SLOPE (Marx & Vreeken, 2017) | - | **81.0%** |
| Our AEQ comparison | - | 80.0% |
| Jarfo (Fonollosa, 2016) | + | 59.5% |
| RCC (Lopez-Paz et al., 2015) | + | 75.0%[b] |
| NCC (Lopez-Paz et al., 2017) | + | 79.0% |

[a]The accuracy of CURE is reported on version 0.8 of the dataset in (Sgouritsa et al., 2015) as 75%. In Bloebaum et al. (2018) they reran this algorithm and achieved an accuracy rate around 60%.

[b]The accuracy scores reported in (Lopez-Paz et al., 2015) are for version 0.8 of the dataset, in (Lopez-Paz et al., 2017) they reran RCC (Lopez-Paz et al., 2015) on version 1.0 of the dataset.

dimensions and their intensity until the autoencoder reconstruction error of both $X$ and $Y$ became similar. Note that for these multivariate variables, we do not use quantiles and use the variables themsevles. As the AutoEncoder reconstruction results in Tab. 3 show, in the MCE-SigMix dataset balancing was only partly successful.

In Tab. 3 we also compare our method to BivariateFit and ANM (Hoyer et al., 2009) algorithms on these datasets and present favorable performance. The other literature methods are designed for the univariate case and therefore, we were unable to extend them. The only baseline methods that are extendable to multi dimensions are PNL (Zhang & Hyvrinen, 2010), CGNN (Goudet et al., 2018) and RECI (Bloebaum et al., 2018). ~~However, their runtime becomes too long. Specifically,~~ However, RECI's runtime is of order $\mathcal{O}(n^3)$, where $n$ is the input dimension.

Table 3: Mean AUC rates of various baselines on different multidimensional dimensional cause-effect pairs datasets. The datasets are designed and balanced, such that an autoencoder method would fail. Our method achieves SOTA results.

| Method | MCE-Poly | MCE-Net | MCE-SigMix | MOUS-MEG |
|---|---|---|---|---|
| BivariateFit | 54.7 | 48.4 | 48.2 | 44.2 |
| ANM (Hoyer et al., 2009) | 52.2 | 51.1 | 46.4 | 52.4 |
| PNL (Zhang & Hyvrinen, 2010) | 76.4 | 54.7 | 16.8 | 56.3 |
| CGNN (Goudet et al., 2018) | 47.8 | 67.8 | 58.8 | 40.9 |
| AE reconstruction | 57.2 | 42.4 | 22.3 | 41.2 |
| Our method | 95.3 | 70.0 | 98.5 | 87.6 |

In addition to the synthetic dataset, we also employ the MOUS-MEG real world dataset, provided to us by the authors of (Anonymous, 2020). This dataset is part of Mother Of Unification Studies (MOUS) dataset (Schoffelen et al., 2019). This dataset contains magneto-encephalography (MEG) recordings of 102 healthy dutch-speaking subjects performing a reading task (9 of them were excluded due to corrupted data). Each subject was asked to read 120 sentences in Dutch, both in the right order and randomly mixed order, which adds up to a total of over 1000 words. Each word was presented on the computer screen for 351ms on average and was separated from the next word by 3-4 seconds. Each time step consists of 301 MEG readings of the magnetometers, attached to different parts of the head. For more information see (Schoffelen et al., 2019). For each pair $(X, Y)$, $X$ is the interval $[-1.5s, -0.5s]$ relative to the word onset concatenated with the word embedding (using the spaCy python module with Dutch language model), this presents the subject in his "rest" state (i.e. the cause). $Y$ is the interval $[0, 1.0s]$ relative to the word onset, which presents the subject in his "active" state (i.e. the effect). The results, also given in Tab. 3 show a clear advantage over the literature methods.

To validate the soundness of the dataset, we ran made a few experiments on variations of the dataset and report the results in Tab. 4. As can be seen, a dataset where the cause consists of the word embedding and the effect consists of the subject's "active" state is highly imbalanced. This is reasonable since the word embedding and the MEG readings are encoded differently and are of different dimensions. In addition, when the cause is selected to be the "rest" state and the effect is the "active" state, the various algorithms are unable to infer which side is the cause and which one is the effect since the word is missing.

**Emergence of independence** To check the importance of our adversarial loss in identifying the direction of causality and capturing the implicit independent representation $f(X)$ and $E$, we applied our method without training $R$ against the discriminator. Therefore, in this case, the discriminator only serves as a test whether $X$ and $R(Y)$ are independent or not. Recall that our analysis shows that the discriminator is not necessary under reasonable conditions. As can be seen in Tab. 5, the adversarial loss improves the results when there is no implicit emergence of independence, however, in cases where there is emergence of independence, the results are similar. ~~This is verified in Tab. 5, which shows similar performance with and without the discriminator.~~

As mentioned in Sec. 3.1, the distance between $c_{\text{real}} + c_{\text{fake}}$ to 1 indicates the amount of dependence between $X$ and $R(Y)$. We denote by Ind C the mean values of $|c_{\text{real}} + c_{\text{fake}} - 1|$ over all pairs of random variables, epochs and samples when training our method in the causal direction. The same mean score when training in the anti-causal direction is denoted Ind E. As is evident from Tab. 5, the independence is similar between the two directions, emphasizing the importance of the reconstruction error in the score.

We noticed that the values of Ind C and Ind E are smaller for the full method. However, in MCE-Poly and MCE-SigMix they are still very small, and therefore, there is implicit emergence of independence between $X$ and $R(Y)$, even without explicitly training $R(Y)$ to be independent of $X$. This can explain the fact that the drop in the AUC is significant but not drastic.

Table 4: Results of various methods on different variations of the MOUS-MEG dataset.

| Method | Dataset | | |
|---|---|---|---|
| | Rest, Word → Active | Rest → Active | Word → Active |
| BivariateFit | 44.2 | 58.1 | 0.0 |
| ANM (Hoyer et al., 2009) | 52.4 | 49.3 | 0.0 |
| AE reconstruction | 41.2 | 51.7 | 98.6 |
| Our method | 87.6 | 44.4 | 0.0 |

Table 5: Emergence of independence. Ind C (Ind E) is the mean of $|c_{\text{real}} + c_{\text{fake}} - 1|$ over all pairs of random variables, epochs and samples, when training the method from $X$ to $Y$ (vice versa). w/o backprop means without backpropagating gradients from $D$ to $R$.

| Dataset | Full method | | | w/o backprop | | |
|---|---|---|---|---|---|---|
| | AUC | Ind C | Ind E | AUC | Ind C | Ind E |
| MCE-Poly | 95.3 | 0.06 | 0.05 | 95.1 | 0.10 | 0.10 |
| MCE-Net | 70.0 | 0.16 | 0.20 | 65.1 | 0.55 | 0.55 |
| MCE-SigMix | 98.5 | 0.05 | 0.06 | 98.8 | 0.16 | 0.20 |
| MOUS-MEG | 87.6 | 0.61 | 0.61 | 80.7 | 0.74 | 0.75 |

## 5 SUMMARY

We identify an inbalance in the complexities of cause and effect in the unidimensional SCM ~~model~~ and suggest a method to exploit it. Since the method does not consider the interactions between the variables, its success in predicting cause and effect indicates an inherent bias in the unidimensional datasets. Turning our attention to the multivariate case, where the complexity can be actively balanced, we propose a new method in which the learned networks models the underlying SCM ~~model~~ itself. Since the noise term $E$ is unknown, we replace it by a function of $Y$ that is enforced to be independent of $X$. We also show that under reasonable conditions, the independence emerges even without explicitly enforcing it.

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

# A  ARCHITECTURE

The function $G$, $F$, $R$ and $D$ in the multivariate method in Sec. 3 are fully connected neural networks and their architectures are as follows: $F$ is a 2-layered network with dimensions $100 \to 60 \to 50$, $R$ is a 3-layered network with dimensions $100 \to 50 \to 50 \to 20$, $G$ is a 2-layers neural network with dimensions $50 + 20 \to 80 \to 100$ (the input has 50 dimensions for $F(X)$ and 20 for $R(Y)$). The discriminator is a 3-layers network with dimensions $100 + 20 \to 60 \to 50 \to 2$ (the input is $X$ and $R(Y)$). The activation function in all networks is the sigmoid function except the discriminator that applies the leaky ReLU activation. For all networks, the activation is not applied at the output layer.

# B  ANALYSIS

## B.1  TERMINOLOGY AND NOTATIONS

We recall some relevant notations and terminology. The identity matrix of dimension $n$ is denoted by $I_n$. For a vector $x = (x_1, \ldots, x_n) \in \mathbb{R}^n$ we denote $\|x\|_2 := \sqrt{\sum_{i=1}^n x_i^2}$ the Euclidean norm of $x$. For a differentiable function $f : \mathbb{R}^m \to \mathbb{R}^n$ and $x \in \mathbb{R}^m$, we denote by

$$\mathrm{J}(f(x)) := \left( \frac{\partial f_i}{\partial \zeta_j}(x) \right)_{i \in [n], j \in [m]} \tag{12}$$

the Jacobian matrix of $f$ in $x$. For a twice differentiable function $f : \mathbb{R}^m \to \mathbb{R}$, we denote by

$$\mathrm{H}(f(x)) := \left( \frac{\partial^2 f}{\partial \zeta_i \partial \zeta_j}(x) \right)_{i, j \in [m]} \tag{13}$$

the Hessian matrix of $f$ in $x$. Additionally, for a twice differentiable function $f : \mathbb{R}^m \to \mathbb{R}^n$, $f(x) = (f_1(x), \ldots, f_n(x))$, we denote the Hessian of $f$ by $\mathrm{H}(f(x)) := (\mathrm{H}(f_1(x)), \ldots, \mathrm{H}(f_n(x)))$. For a scalar function $f : \mathbb{R}^m \to \mathbb{R}$ instead of using the Jacobian notation, the gradient notation will be employed, $\nabla(f(x)) := \mathrm{J}(f(x))$.

## B.2  AUTOENCODER AS A COMPLEXITY MEASURE

Let $\mathcal{F} = \{\mathcal{H}^d\}_{d=1}^\infty$ be a family of classes of autoencoders $A : \mathbb{R}^d \to \mathbb{R}^d$. Assume that the family $\mathcal{F}$ is closed to fixations, i.e., for any autoencoder $A \in \mathcal{H}^{d_1 + d_2}$ and any fixed vector $y^* \in \mathbb{R}^{d_2}$ ($x^* \in \mathbb{R}^{d_1}$), we have: $A(x, y^*)_{1:d_1} \in \mathcal{H}^{d_1}$ ($A(x^*, y)_{d_1+1:d_2} \in \mathcal{H}^{d_2}$). Here, $v_{i:j} = (v_i, \ldots, v_j)$. We consider that this is the typical situation when considering neural networks with biases.

Let $X$ be a random variable. Let $X$ be a multivariate random variable dimension $d$. We define the autoencoding complexity of $X$ as follows:

$$C(X; \mathcal{F}) := \min_{A^* \in \mathbb{H}^d} \mathbb{E}_{x \sim X} \left[ \ell(A^*(x), x) \right] \tag{14}$$

**Lemma 3.** *Let $\{\mathcal{H}^d\}_{d=1}^\infty$ be a family of classes of autoencoders that is closed to fixations. The function $C(X; \mathcal{F})$ is a complexity measure according to Sec. 3.*

*Proof.* First, since $\ell(a, b) \geq 0$ for all $a, b \in \mathbb{R}^k$, this function is non-negative. Next, we would like to show that $C^\alpha(X, Y) \geq \max(C(X), C(Y))$. Let $A^*$ be the minimizer of $\mathbb{E}_{x \sim X} \left[ \ell(A^*(x), x) \right]$ within $\mathcal{H}^{d_1 + d_2}$. We consider that there is a vector $y^*$, such that,

$$\begin{aligned} \mathbb{E}_{(x,y) \sim (X,Y)} \left[ \ell(A(x,y), (x,y)) \right] &\geq \mathbb{E}_{y \sim Y} \mathbb{E}_{x \sim X} \left[ \ell(A(x,y), (x,y)) \right] \\ &\geq \mathbb{E}_{x \sim X} \left[ \ell(A(x, y^*), (x, y^*)) \right] \\ \geq \mathbb{E}_{x \sim X} \left[ \ell(A(x, y^*)_{1:d_1}, x) \right] \end{aligned} \tag{15}$$

We note that $A(x, y^*)_{1:d_1} \in \mathcal{H}^{d_1}$. Therefore,

$$\mathbb{E}_{(x,y) \sim (X,Y)} \left[ \ell(A(x,y), (x,y)) \right] \geq \min_{A^* \in \mathcal{H}^{d_1}} \mathbb{E}_{x \sim X} \left[ \ell(A^*(x), x) \right] = C(X; \mathcal{F}) \tag{16}$$

By similar considerations, $C(X, Y; \mathcal{F})$. □

### B.3 PROOFS FOR THE RESULTS

In this section we provide the proofs of the main results in the paper.

**Lemma 1.** *In the setting of Sec. 2.2. Let $\theta^*$ be the global optimum of Eq. 4 for a sufficiently large $\mathcal{F}$. Assume that $\mathcal{P}$ is sufficiently large, such that:*

$$\forall z \in \mathcal{Z}, \phi \in \Phi : q_\phi(u|z) \in \mathcal{P} \tag{5}$$

*Then,*
$$
\begin{aligned}
h(U) =& \frac{1}{2\sigma_1^2} \cdot \mathbb{E}_{u \sim U} \left[ \mathbb{E}_{\epsilon \sim \mathcal{N}(0, I_d)} \left[ \| f_{\theta^*}(g_\phi(u) + \sigma_2^2 \cdot I_d \cdot \epsilon) - u \|_2^2 \right] \right] \\
&+ \frac{1}{2} \mathbb{E}_{u \sim U} \left[ \| g_\phi(u) \|_2^2 \right] + \frac{d \cdot (\sigma_2^2 - 1 - \log(\sigma_2^2))}{2} + \frac{n}{2} \log(2\pi\sigma_1^2)
\end{aligned}
\tag{6}
$$

*Proof.* This is a corollary of the analysis of Zhao et al. (2017). By their analysis, we have:

$$h(U) = I_{q_\phi}(z; u) - \mathbb{E}_{q_\phi} \left[ \log p_{\theta^*}(u|z) \right] \tag{17}$$

Since we selected $p_{\theta^*}(u|z)$ to be the PDF of a normal distribution $\mathcal{N}(f_{\theta^*}(z), \sigma_1^2 \cdot I_n)$, we have:

$$\log p_{\theta^*}(u|z) = -\frac{\| f_{\theta^*}(z) - u \|_2^2}{2\sigma_1^2} - \frac{n}{2} \log(2\pi\sigma_1^2) \tag{18}$$

Therefore, since $z$ distributed by $q_\phi(z|u)$ can be represented as $z|u \sim g_\phi(u) + \sigma_2^2 \cdot I_d \cdot \epsilon$, where $\epsilon \sim \mathcal{N}(0, I_d)$,

$$
\begin{aligned}
-\mathbb{E}_{q_\phi(u,z)} \left[ \log p_{\theta^*}(u|z) \right] =& \frac{1}{2\sigma_1^2} \cdot \mathbb{E}_{q_\phi(u,z)} \left[ \| f_{\theta^*}(z) - u \|_2^2 \right] + \frac{n}{2} \log(2\pi\sigma_1^2) \\
=& \frac{1}{2\sigma_1^2} \cdot \mathbb{E}_{u \sim U} \left[ \mathbb{E}_{q_\phi(z|u)} \left[ \| f_{\theta^*}(z) - u \|_2^2 \right] \right] + \frac{n}{2} \log(2\pi\sigma_1^2) \\
=& \frac{1}{2\sigma_1^2} \cdot \mathbb{E}_{u \sim U} \left[ \mathbb{E}_{\epsilon \sim \mathcal{N}(0, I_d)} \left[ \| f_{\theta^*}(g_\phi(u) + \sigma_2^2 \cdot I_d \cdot \epsilon) - u \|_2^2 \right] \right] \\
&+ \frac{n}{2} \log(2\pi\sigma_1^2)
\end{aligned}
\tag{19}
$$

In addition, we have:

$$
\begin{aligned}
I_{q_\phi}(z; u) &= \mathbb{E}_{u \sim U} [D_{\mathrm{KL}}(q_\phi(z|u) \| p(z))] \\
&= \mathbb{E}_{u \sim U} [D_{\mathrm{KL}}(\mathcal{N}(g_\phi(u), \sigma_2^2 \cdot I_d) \| \mathcal{N}(0, I_d))] \\
&= \mathbb{E}_{u \sim U} \left[ \frac{1}{2} \left( d \cdot \sigma_2^2 + \| g_\phi(u) \|_2^2 + d \log(\sigma_2^2) \right) \right] \\
&= \frac{d \cdot (\sigma_2^2 - 1 - \log(\sigma_2^2))}{2} + \frac{1}{2} \mathbb{E}_u \left[ \| g_\phi(u) \|_2^2 \right]
\end{aligned}
\tag{20}
$$

Therefore, we obtained the desired equation. □

**Lemma 2.** *Let $C$ be a complexity measure of multivariate random variables (i.e, non-negative and satisfies Eq. 7). Then, there are triplets of random variables $(X, E, Y)$ and $(\hat{X}, E, Y)$, such that, $Y = g(X, E)$, $Y = g'(\hat{X}, E)$, $C(X) < C(Y)$ and $C(\hat{X}) > C(Y)$. Therefore, $C$ cannot serve as a score for causal inference.*

*Proof.* Let $X$ be a random variable and $E \perp\!\!\!\perp X$, such that, $Y = g(X, E)$. Assume that $C(X) < C(Y)$. Then, let $X'$ be a random variable independent of $X$, such that, $C(X') > C(Y)$. Then, we have: $C(X, X') > C(Y)$ and we have: $Y = g'(X, X', E)$, for $g'(a, b, c) = g(a, c)$. □

The following lemma is an extension of Thm. 1 in (Zhang & Hyvrinen, 2010) to real valued random variables of dimension $> 1$.

**Lemma 4.** *Assume that $(X, Y)$ can be described by both:*

$$Y = g_1(f_1(X) + E_1), \text{ s.t: } X \perp\!\!\!\perp E_1 \text{ and } g_1 \text{ is invertible} \tag{21}$$

*and*

$$X = g_2(f_2(Y) + E_2), \text{ s.t: } Y \perp\!\!\!\perp E_2 \text{ and } g_2 \text{ is invertible} \tag{22}$$

*Assume that $g_1$ and $g_2$ are invertible and let:*

$$
\begin{aligned}
T_1 &:= g_1^{-1}(Y) \text{ and } h_1 := f_2 \circ g_1 \\
T_2 &:= g_2^{-1}(X) \text{ and } h_2 := f_1 \circ g_2
\end{aligned}
\tag{23}
$$

*Assume that the involved densities $p_{T_2}$, $p_{E_1}$ and nonlinear functions $f_1, g_1$ and $f_2, g_2$ are third order differentiable. We then have the following equations for all $(X, Y)$ satisfying:*

$$
\begin{aligned}
&\mathbf{H}(\eta_1(t_2)) \cdot \mathbf{J}(h_1(t_1)) - \mathbf{H}(\eta_2(e_1)) \cdot \mathbf{J}(h_2(t_2)) \\
&+ \mathbf{H}(\eta_2(e_1)) \cdot \mathbf{J}(h_2(t_2)) \cdot \mathbf{J}(h_1(t_1)) \cdot \mathbf{J}(h_2(t_2)) \\
&- \nabla(\eta_2(e_1)) \cdot \mathbf{H}(h_2(t_2)) \cdot \mathbf{J}(h_1(t_1)) = 0
\end{aligned}
\tag{24}
$$

*where $\eta_1(t_2) := \log p_{T_2}(t_2)$ and $\eta_2(e_1) := \log p_{E_1}(e_1)$.*

*Proof.* The proof is an extension of the proof of Thm. 1 in (Zhang & Hyvrinen, 2010). We define:

$$
\begin{aligned}
T_1 &:= g_1^{-1}(Y) \text{ and } h_1 := f_2 \circ g_1 \\
T_2 &:= g_2^{-1}(X) \text{ and } h_2 := f_1 \circ g_2
\end{aligned}
\tag{25}
$$

Since $g_2$ is invertible, the independence between $X$ and $E_1$ is equivalent to the independence between $T_2$ and $E_1$. Similarly, the independence between $Y$ and $E_2$ is equivalent to the independence between $T_1$ and $E_2$. Consider the transformation $F : (E_2, T_1) \mapsto (E_1, T_2)$:

$$
\begin{aligned}
E_1 &= T_1 - f_1(X) = T_1 - f_1(g_2(T_2)) \\
T_2 &= f_2(Y) + E_2 = f_2(g_1(T_1)) + E_2
\end{aligned}
\tag{26}
$$

The Jacobian matrix of this transformation is given by:

$$\mathbf{J} := \mathbf{J}(F(e_2, t_1)) = \left[ \begin{array}{c|c} -\mathbf{J}(h_2(t_2)) & I - \mathbf{J}(h_2(t_2)) \cdot \mathbf{J}(h_1(t_1)) \\ \hline I & \mathbf{J}(h_1(t_1)) \end{array} \right] \tag{27}$$

Since $I$ commutes with any matrix, by Thm. 3 in (Silvester, 1999), we have:

$$\left| \det(\mathbf{J}(F(E_2, T_1))) \right| = \left| \det \left( -\mathbf{J}(h_2(T_2)) \cdot \mathbf{J}(h_1(T_1)) - I \cdot (I - \mathbf{J}(h_2(T_2)) \cdot \mathbf{J}(h_1(T_1))) \right) \right| = 1 \tag{28}$$

Therefore, we have: $p_{T_2}(t_2) \cdot p_{E_1}(e_1) = p_{T_1, E_2}(t_1, e_2) / |\det \mathbf{J}| = p_{T_1, E_2}(t_1, e_2)$. Hence, $\log(p_{T_1, E_2}(t_1, e_2)) = \eta_1(t_2) + \eta_2(e_1)$ and we have:

$$\frac{\partial \log(p_{T_1, E_2}(t_1, e_2))}{\partial e_2} = \nabla \eta_1(t_2) - \nabla \eta_2(e_1) \cdot \mathbf{J}(h_2(t_2)) \tag{29}$$

Therefore,

$$
\begin{aligned}
\frac{\partial^2 \log(p_{T_1, E_2}(t_1, e_2))}{\partial e_2 \partial t_1} =& \mathbf{H}(\eta_1(t_2)) \cdot \mathbf{J}(h_1(t_1)) - \mathbf{H}(\eta_2(e_1)) \cdot (I - \mathbf{J}(h_2(t_2)) \cdot \mathbf{J}(h_1(t_1))) \cdot \mathbf{J}(h_2(t_2)) \\
& - \nabla(\eta_2(e_1)) \cdot \mathbf{H}(h_2(t_2)) \cdot \mathbf{J}(h_1(t_1)) \\
=& \mathbf{H}(\eta_1(t_2)) \cdot \mathbf{J}(h_1(t_1)) - \mathbf{H}(\eta_2(e_1)) \cdot \mathbf{J}(h_2(t_2)) \\
& + \mathbf{H}(\eta_2(e_1)) \cdot \mathbf{J}(h_2(t_2)) \cdot \mathbf{J}(h_1(t_1)) \cdot \mathbf{J}(h_2(t_2)) \\
& - \nabla(\eta_2(e_1)) \cdot \mathbf{H}(h_2(t_2)) \cdot \mathbf{J}(h_1(t_1))
\end{aligned}
\tag{30}
$$

The independence between $T_1$ and $E_2$ implies that for every possible $(t_1, e_2)$, we have: $\frac{\partial^2 \log p_{T_1, E_2}(t_1, e_2)}{\partial e_2 \partial t_1} = 0$. $\qquad \square$

**Lemma 5** (Reduction to post-linear models). *Let $f(x) = \sigma_1(W_d \ldots \sigma_1(W_1 x))$ and $g(u, v) = \sigma_2(U_k \ldots \sigma_2(U_1(u, v)))$ be two neural networks. Then, if $Y = g(f(X), E)$ for some $E \perp\!\!\!\perp X$, we can represent $Y = \hat{g}(\hat{f}(X) + N)$ for some $N \perp\!\!\!\perp X$.*

*Proof.* Let $f(x) = \sigma_1(W_d \ldots \sigma_1(W_1 x))$ and $g(u, v) = \sigma_2(U_k \ldots \sigma_2(U_1(u, v)))$ be two neural networks. Here, $(u, v)$ is the concatenation of the vectors $u$ and $v$. We consider that $U_k(f(X), E) = U_1^1 f(X) + U_1^2 E$. We define a noise variable $N := U_1^2 E$ and have: $X \perp\!\!\!\perp N$. In addition, let $\hat{f}(x) := U_1^1 f(x)$ and $\hat{g}(z) := \sigma_2(U_k \ldots \sigma_2(U_2 \sigma_2(z)))$. We consider that: $Y = \hat{g}(\hat{f}(X) + N)$ as desired. $\square$

**Theorem 1** (Identifiability of neural SCMs). *Let $\mathbb{P}_{X,Y}$ admit a neural SCM from $X$ to $Y$ as in Eq. 1, such that $p_X$, and the activation functions of $f$ and $g$ are three-times differentiable. Then it admits a neural SCM from $Y$ to $X$ only if $p_X$, $f$, $g$ satisfy Eq. 34 in the Appendix.*

*Proof.* Let $f_i(z) = \sigma_1(W_{i,d} \ldots \sigma_1(W_{i,1} z))$ and $g_i(u, v) = \sigma_2(U_{i,k} \ldots \sigma_2(U_{i,1}(u, v)))$ (where $i = 1, 2$) be pairs of neural networks, such that, $\sigma_1$ and $\sigma_2$ are three-times differentiable. Assume that:

$$Y = g(f(X), E_1) \text{ and } X = g(f(Y), E_2) \tag{31}$$

for some $E_1 \perp\!\!\!\perp X$ and $E_2 \perp\!\!\!\perp Y$. By Lem. 5, we can represent

$$Y = \hat{g}_1(\hat{f}_1(X) + N_1),$$
$$\text{where } N_1 = U_{1,1}^2 E_1, \ \hat{f}_1 = U_{1,1}^1 f_1(X) \text{ and } \hat{g}_1(z) = \sigma_2(U_{1,k} \ldots \sigma_2(U_{1,2} \sigma_2(z))) \tag{32}$$

and also,

$$X = \hat{g}_2(\hat{f}_2(Y) + N_2),$$
$$\text{where } N_2 = U_{2,1}^2 E_2, \ \hat{f}_2 = U_{2,1}^1 f_2(X) \text{ and } \hat{g}_2(z) = \sigma_2(U_{2,k} \ldots \sigma_2(U_{2,2} \sigma_2(z))) \tag{33}$$

From the proof of Lem. 5, it is evident that the constructed $\hat{g}_1$, $\hat{f}_1$ and $\hat{g}_1$, $\hat{f}_2$ are three-times differentiable whenever $\sigma_1$ and $\sigma_2$ are. Therefore, by Thm. 1, the following differential equation holds:

$$\begin{aligned} &\mathbf{H}(\eta_1(t_2)) \cdot \mathbf{J}(h_1(t_1)) - \mathbf{H}(\eta_2(n_1)) \cdot \mathbf{J}(h_2(t_2)) \\ &+ \mathbf{H}(\eta_2(n_1)) \cdot \mathbf{J}(h_2(t_2)) \cdot \mathbf{J}(h_1(t_1)) \cdot \mathbf{J}(h_2(t_2)) \\ &- \nabla(\eta_2(n_1)) \cdot \mathbf{H}(h_2(t_2)) \cdot \mathbf{J}(h_1(t_1)) = 0 \end{aligned} \tag{34}$$

where

$$\begin{aligned} T_1 &:= \hat{g}_1^{-1}(Y) \text{ and } h_1 := \hat{f}_2 \circ \hat{g}_1 \\ T_2 &:= \hat{g}_2^{-1}(X) \text{ and } h_2 := \hat{f}_1 \circ \hat{g}_2 \end{aligned} \tag{35}$$

and $\eta_1(t_2) := \log p_{T_2}(t_2)$ and $\eta_2(n_1) := \log p_{N_1}(n_1)$. $\square$

**Theorem 2** (Uniqueness of Representation). *Let $\mathbb{P}_{X,Y}$ admit a nonlinear model from $X$ to $Y$ as in Eq. 1, i.e., $Y = g(f(X), E)$ for some random variable $E \perp\!\!\!\perp X$. Assume that $f$ and $g$ are invertible. Let $G$, $F$ and $R$ be functions, such that, $\mathcal{L}_{err} := \mathbb{E}_{(x,y) \sim (X,Y)}[\|G(F(x), R(y)) - y\|_2^2] = 0$ and $G$ and $F$ are invertible functions and $X \perp\!\!\!\perp R(Y)$. Then, $F(X) \propto f(X)$ and $R(Y) \propto E$.*

*Proof.* Since $F$ and $f$ are invertible, one can represent: $F(X) = F(f^{-1}(f(X)))$ and $f(X) = f(F^{-1}(F(X)))$. Similarly, since $G$ and $g$ are invertible, we also have: $(F(X), R(Y)) \propto (f(X), E)$. Since $(F(X), R(Y)) \propto (f(X), E)$ and $F(X) \propto f(X)$, we have: $R(Y) = Q(F(X), E)$. However, $R(Y) \perp\!\!\!\perp F(X)$ and therefore, we can represent $R(Y) = P(E)$ and vice versa. $\square$

**Theorem 3** (Emergence of independent representations). *Let $\mathbb{P}_{X,Y}$ admits a nonlinear model from $X$ to $Y$ as in Eq. 1, i.e., $Y = g(f(X), E)$ for some random variable $E \perp\!\!\!\perp X$. Assume that $X$ and $Y$ are discrete random variables. Let $G$, $F$ and $R$ be functions, such that, $\mathcal{L}_{err} := \mathbb{E}_{(x,y) \sim (X,Y)}[\|G(F(x), R(y)) - y\|_2^2] = 0$ and $G$ is an invertible function. Assume that $h(R(Y)) \leq h(E)$ and that $g$ is invertible. Then, we have: $F(X) \perp\!\!\!\perp R(Y)$, $F(X) \propto f(X)$ and $R(Y) \propto E$.*

*Proof.* Since $g$ and $f$ are invertible, we have: $h(Y) = h(f(X)) + h(E) \leq h(X) + h(E)$. In addition, we have: $h(R(Y)) \leq h(E)$. Therefore, since $G$ is invertible, $h(Y) = h(F(X), R(Y)) \leq h(F(X)) + h(R(Y)) \leq h(X) + h(E) = h(Y)$. Hence, $F(X)$ and $R(Y)$ are independent. □

In the following three lemmas we extend Thms. 2 and 3.

**Lemma 6.** *Let $\mathbb{P}_{X,Y}$ admit a nonlinear model from $X$ to $Y$ as in Eq. 1, i.e., $Y = g(f(X), E)$, for some random variable $E \perp\!\!\!\perp X$. Assume that $f$ and $g$ are invertible. Assume that $X$ and $Y$ are discrete and $Y$ is over a set $\mathcal{Y}$, such that, $\forall y_1 \neq y_2 \in \mathcal{Y}$, we have, $\|y_1 - y_2\|_2 > \Delta > 0$. Let $G$, $F$ and $R$ be functions, such that, $\mathcal{L}_{\text{err}} := \mathbb{E}_{(x,y) \sim (X,Y)} \left[ \|y - G(F(x), R(y))\|_2^2 \right] < \Delta/2$ and $F$ is invertible. Then, $F(X) \propto f(X)$ and $I(R(Y); E) \geq \left(1 - \frac{\mathcal{L}_{\text{err}}}{\Delta}\right) \cdot h(E) - \sqrt{\frac{\mathcal{L}_{\text{err}}}{\Delta}}$.*

*Proof.* First, since both $F$ and $f$ are invertible, $F(X) \propto f(X)$. By the proof of Lem. 10 in (Press et al., 2019), there is a function $r$, such that:

$$\mathbb{P}_{(x,y) \sim (X,Y)}[r(F(x), R(y)) = y] \geq \left(1 - \frac{\mathcal{L}_{\text{err}}}{\Delta}\right) \geq 0.5 \tag{36}$$

Since $g$ is invertible, there is a function $u(Y) = E$. In particular,

$$\mathbb{P}_{(x,y) \sim (X,Y)}[u(r(F(x), R(y))) = u(y)] \geq \left(1 - \frac{\mathcal{L}_{\text{err}}}{\Delta}\right) \geq 0.5 \tag{37}$$

Therefore, by Lem. 6 in (Press et al., 2019), we have:

$$I(F(X), R(Y); E) \geq \left(1 - \frac{\mathcal{L}_{\text{err}}}{\Delta}\right) \cdot h(E) - h\left(1 - \frac{\mathcal{L}_{\text{err}}}{\Delta}\right) \tag{38}$$

By the analysis in the proof of Lem. 10 in (Press et al., 2019), we derive that:

$$I(F(X), R(Y); E) \geq \left(1 - \frac{\mathcal{L}_{\text{err}}}{\Delta}\right) \cdot h(E) - \sqrt{\frac{\mathcal{L}_{\text{err}}}{\Delta}} \tag{39}$$

Finally, since $F(X)$ is a function of $X$ which is independent of $E$, we obtain the desired inequality:

$$I(R(Y); E) \geq \left(1 - \frac{\mathcal{L}_{\text{err}}}{\Delta}\right) \cdot h(E) - \sqrt{\frac{\mathcal{L}_{\text{err}}}{\Delta}} \tag{40}$$

□

We mention that in Lem. 6, the function $R(Y)$ can hold all of the information present in $Y$. Therefore, in order to suffice that $R(Y)$ holds only the information present in $E$, one can restrict that $h(R(Y)) \leq h(E) + \epsilon$ as will be shown in the next lemma. We note that under the conditions of Lem. 6, $h(R(Y)) \leq h(E) + \epsilon_1$ is equivalent to $I(F(X); R(Y)) \leq \epsilon_2$ for $\epsilon_1, \epsilon_2$ that are functions of each other.

**Lemma 7.** *In the setting of Lem. 6. Assume that $h(R(Y)) \leq h(E) + \epsilon$, for some constant $\epsilon > 0$. Then, $I(F(X); R(Y)) \leq \frac{\mathcal{L}_{\text{err}}}{\Delta} \cdot h(Y) + \sqrt{\frac{\mathcal{L}_{\text{err}}}{\Delta}} + \epsilon$. In addition, there are functions $r_1, r_2$, such that,*

$$\mathbb{P}[r_1(R(Y)) \neq E] \leq 1 - 2^{-\mathcal{L}_{\text{err}} \cdot h(E)/\Delta - \sqrt{\mathcal{L}_{\text{err}}/\Delta}} \tag{41}$$

*and*

$$\mathbb{P}[r_2(E) \neq R(Y)] \leq 1 - 2^{-\mathcal{L}_{\text{err}} \cdot h(E)/\Delta - \sqrt{\mathcal{L}_{\text{err}}/\Delta} - \epsilon} \tag{42}$$

*Proof.* By Lem. 10 in Press et al. (2019), we have:

$$h(F(X), R(Y)) \geq h(F(X), R(Y); Y) \geq \left(1 - \frac{\mathcal{L}_{\text{err}}}{\Delta}\right) \cdot h(Y) - \sqrt{\frac{\mathcal{L}_{\text{err}}}{\Delta}} \tag{43}$$

In addition, by the data processing inequality and the assumption $h(R(Y)) \leq h(E) + \epsilon$, we have:

$$
\begin{aligned}
I(F(X); R(Y)) &\leq h(F(X)) + h(R(Y)) - h(F(X), R(Y)) \\
&\leq h(X) + h(Y) + \epsilon - \left( \left( 1 - \frac{\mathcal{L}_{\text{err}}}{\Delta} \right) \cdot h(Y) - \sqrt{\frac{\mathcal{L}_{\text{err}}}{\Delta}} \right) \\
&\leq h(Y) + \epsilon - \left( \left( 1 - \frac{\mathcal{L}_{\text{err}}}{\Delta} \right) \cdot h(Y) - \sqrt{\frac{\mathcal{L}_{\text{err}}}{\Delta}} \right) \\
&\leq \frac{\mathcal{L}_{\text{err}}}{\Delta} \cdot h(Y) + \sqrt{\frac{\mathcal{L}_{\text{err}}}{\Delta}} + \epsilon
\end{aligned}
\tag{44}
$$

Finally, by Feder & Merhav (1994), there is a function $r_1$, such that,

$$
\begin{aligned}
\mathbb{P}[r_1(R(Y)) \neq E] &\leq 1 - 2^{I(R(Y);E) - h(E)} \\
&\leq 1 - 2^{-\mathcal{L}_{\text{err}} \cdot h(E)/\Delta - \sqrt{\mathcal{L}_{\text{err}}/\Delta}}
\end{aligned}
\tag{45}
$$

In addition, there is a function $r_2$, such that,

$$
\begin{aligned}
\mathbb{P}[r_2(E) \neq R(Y)] &\leq 1 - 2^{I(R(Y);E) - h(R(Y))} \\
&\leq 1 - 2^{I(R(Y);E) - h(E) - \epsilon} \\
&\leq 1 - 2^{-\mathcal{L}_{\text{err}} \cdot h(E)/\Delta - \sqrt{\mathcal{L}_{\text{err}}/\Delta} - \epsilon}
\end{aligned}
\tag{46}
$$

as desired. $\qquad\square$

