# OpenReview forum: "Variable Complexity in the Univariate and Multivariate Structural Causal Model"
_ICLR.cc/2020/Conference — Reject_

### Official Review · AnonReviewer2 · 2019-10-23
**Official Blind Review #2**

**Rating:** 6

**Review:**

Update:

The authors have successfully justified my concerns. Therefore, I have increased my score to 6.

Original comments:

In this paper, the authors consider learning causal directions from observational data from both univariate case and multi-dimensional case. In the univariate case, the authors propose a new method to learn causal directions by exploiting the complexities of cause and effect variables. In the multi-dimensional case, where the complexity can be balanced, the authors proposed a method that learns causal direction based on independence loss.

1. The independence loss part looks confusing to me. Standard results in SCM yields that the error term E is a function of both the outcome Y and X. How can you learn the term E just from Y itself? In other words, I am not sure if the conditions required in Theorem 2 is feasible. The authors need to provide some examples to justify that the conditions in Theorem 2 are feasible conditions.

2. In fact, the novel idea of learning causal directions based on independence test has been extensively studied in the previous literature. I regret that this has not been mentioned in the current manuscript. Examples include:

http://www.jmlr.org/papers/v12/shimizu11a.html

In conclusion, since the idea of using independence relations for learning the causal directions is not a very new idea and a lot of discussion of the theoretical analysis is still missing. I regret that this work seems not strong enough to be accepted by ICLR.

**Experience Assessment:**

I have published one or two papers in this area.

**Review Assessment: Checking Correctness Of Derivations And Theory:**

I carefully checked the derivations and theory.

**Review Assessment: Checking Correctness Of Experiments:**

I assessed the sensibility of the experiments.

**Review Assessment: Thoroughness In Paper Reading:**

I read the paper at least twice and used my best judgement in assessing the paper.

---

> ### Author Response · Authors · 2019-11-13
> **Thank you for the insightful comments**
>
> Univariate case: please see the general comments.
>
> 1. The common assumption in the SCM literature is that a cause-effect pair X -> Y can be modeled as Y = g(X,E), where, E is a noise term independent of X (see (cf. Peters et al. 2017, p. 8), Zhang & Hyvarinen (2009), Hoyer et al. (2009), Shimizu et al. (2006)). In theorem 2, we assume that Y is invertible, and therefore, the information of E in encoded within Y. We believe that this is not a strong assumption. Informally, if “parts” of E’s information were not encoded within Y, they could be ignored and we could write Y = g(f(X),E’), where E’ is encoded in Y.
>
> 2. In the related work section, we discuss various methods that apply different independence tests, such as, ANM and LiNGAM. We added PNL and Direct-LiNGAM as well, emphasizing the reliance of these methods on independence tests and how our method differs.
>
> In general, we are aware that independence tests are the obvious thing to do. The novelty of our algorithm stems from applying a discriminator in order to measure and restrict independence.
> There are several advantages in employing a discriminator:
> a. Our loss is non-parametric. We can learn a random variable E = R(Y) independent of X without any explicit assumption on the densities of X,Y or E.
> b. Our method does not assume any specific structure as done by LiNGAM, Direct-LiNGAM, PNL, ANM, etc. In fact, we can provably recover the direction and structure of the SCM (up to transformations), under the assumption that g is invertible. In previous publications, this is possible only under the assumption that the SCM is linear/post-linear (PNL, LiNGAM, ANM, etc’).
> c. We do not rely on estimating and minimizing the mutual information between arguments. It is often hard to measure and optimize the mutual information directly, especially in higher dimensions.

---

### Official Review · AnonReviewer1 · 2019-10-26
**Official Blind Review #1**

**Rating:** 3

**Review:**

This paper proposes to use the autoencoder to measure the complexity for identifying the cause and effect in the univariate case. For the multivariate case, this paper extends the PNL model and use GAN for enforcing the independence between cause and noise.

- However, my main concerns are regarding the assumption of this work, seeing that the assumption h(X)>h(Y) in the univariate case is easy to violate. For example, let Y=f(X)+N (a special case of PNL) with some high entropy N, then h(Y) could higher than the h(X).

- In the multivariate case, it can be seen as incremental for PNL but does not offer new insights.

**Experience Assessment:**

I have published in this field for several years.

**Review Assessment: Checking Correctness Of Derivations And Theory:**

I carefully checked the derivations and theory.

**Review Assessment: Checking Correctness Of Experiments:**

I assessed the sensibility of the experiments.

**Review Assessment: Thoroughness In Paper Reading:**

I read the paper thoroughly.

---

> ### Author Response · Authors · 2019-11-13
> **Thank you for the insightful comments**
>
> 1. Univariate case: please see the general comments.
>
> 2. Regarding the multivariate case, our method differs in several aspects. In PNL, the authors learn a mapping between X and Y (and vice versa) of the form Y = g(f(X) + E), where f(X) and E are restricted to be independent. In order to learn f and g, the algorithm restricts that f(X) and E = g^{-1}(Y) - f(X) to be independent.
>
> Our algorithm solves a few disadvantages of PNL:
> a. Their model strongly relies on the assumption that Y has the form: g(f(X) + E) and therefore, they cannot treat the general case where Y = g(f(X),E) as we do.
> b. In addition, we show theoretically in Thms. 1 and 2 that when applying our method one can recover the direction and structure of the SCM (up to transformations) under the assumption that g and f are invertible. In PNL, this is possible only under the assumption that Y = g(f(X)+E).
> c. In order to restrict the components f(X) and N = g^{-1}(Y) - f(X) (a function of Y and X) to be independent, PNL minimizes the mutual information I(f(X);E). For this purpose, the PNL algorithm computes the gradient of I(f(X);g^{-1}(Y) - f(X)) with respect to the parameters of f and g. This is a strong disadvantage of that method since it is often hard to measure and optimize the mutual information directly, especially in higher dimensions. In most cases, it requires having explicit modeling of the density functions of X and Y (p_X(x) and p_Y(y)). In our method, the independence constraint is applied on the observations rather on explicit modeling of the density functions.
>
> We added a paragraph in the related work section discussing PNL and how our algorithm resolves the above problems.
>
> Finally, for completeness, we added an empirical comparison between our method and a multivariate extension of PNL.

---

### Official Review · AnonReviewer4 · 2019-11-08
**Official Blind Review #3**

**Rating:** 6

**Review:**

Edit after author rebuttal and author additions:

I have updated my score from a weak reject (3) to a weak accept (6).

Justification:

1. The authors have pointed out that I misunderstood one of their contributions: pointing out that they are demonstrating that the univariate case is an insufficient setting to test causal discovery methods because it can be done without even looking at the conditional distributions (just the marginals). This contribution seems important to orient future work. They have made this more clear in their recent upload and would likely make it even more clear in a camera-ready version.

2. The authors have made their contribution to the multivariate setting more substantial by adding evaluation on the MOUS-MEG real-world dataset and have better positioned their work relative to others by adding comparisions to multivariate extensions of PNL and CGNN.

====================================================================================================

Original Review:

Summary: The authors focus on the problem of inferring whether the causal structure X —> Y or Y —> X. They first consider the case where both X and Y are scalar (univariate) random variables and then consider the case where X and Y are vector-valued (multivariate) random variables. In the scalar case, motivated by the idea that the effect could be less entropic than the cause (due to data processing inequality), they introduce a method based on comparing reconstruction losses of X and Y and show competitive results in Tables 1 and 2. They establish that this method is not sufficient for the multivariate case in Lemma 2 and move to a new method for the multivariate case. They prove identifiability for this new method in for the multivariate case in Section 4.2 and claim state-of-the-art (SOTA) results in Table 3.

Main contributions:
- Presents a causal discovery technique for the univariate cases that only examines the marginal distributions of X and Y and seems fairly competitive (Tables 1 and 2)
- Extends the post-nonlinear identifiability analysis of Zhang & Hyv¨arinen (2009) from scalars to vectors and proved that their method will actually identify the correct causal direction
- Demonstrates competitive experimental results for both their univariate method
- Claims SOTA results for their multivariate method

Decision: I lean toward rejecting this paper because 1) I have several questions about the univariate case (see below) that would need to be resolved before I lean toward accept, 2) although I am not too familiar with the literature, I believe that this paper may be missing key related work that also uses independence testing for causal discovery (see, e.g., Heinze-Deml et al. (2017)’s Invariant Causal Prediction for Nonlinear Models), and 3) I am not yet convinced that the comparison done in Table 3 is fair and exhaustive.

Sufficient reason to accept: If the theorems in Section 4.2 are found to checkout, and the SOTA results in Table 3 are found to be fair, exhaustive comparisons to the previous SOTA, their contribution to the multivariate case would seem to be sufficient for acceptance. I believe more discussion between the authors and reviewers is necessary here.

 Questions about univariate case:

 1. The motivation for the first method (entropy decreasing along a Markov chain due the data processing inequality) seems to only be valid when Y := f(X), but not necessarily when Y := f(X) + E. For example, let f be the identity function and E be independent to X. How did you resolve this argument against the intuition?

2. Also, I thought the data processing inequality relates mutual information between variables, not necessarily their entropies. Can you make this connection more clear?

Context for questions 3 and 4: In Section 3.1, you write, “estimating the entropy of each random variable from its samples does not present a consistent difference between the entropies h(X)  and h(Y). Our method, therefore, computes an alternative complexity score for X and, independently, for Y.” You then go on to link the entropy to the reconstruction error (your method) in Lemma 1 and show competitive results in Tables 1 and 2.

3. Why do you want to link the reconstruction error to entropy if you found a purely entropy-based method did not work?

4. Why did the purely entropy-based method not work while your method worked if the two are linked?

Questions about multivariate case:

5. Are you certain that BivariateFit and ANM are the only models that you should be comparing against for this multivariate setting?

6. What is CGNN’s runtime? Would you be able to compare against CGNN in time for a potential camera-ready version of this paper?

**Experience Assessment:**

I do not know much about this area.

**Review Assessment: Checking Correctness Of Derivations And Theory:**

I did not assess the derivations or theory.

**Review Assessment: Checking Correctness Of Experiments:**

I assessed the sensibility of the experiments.

**Review Assessment: Thoroughness In Paper Reading:**

I read the paper at least twice and used my best judgement in assessing the paper.

---

> ### Author Response · Authors · 2019-11-13
> **Thank you for the insightful comments**
>
> Regarding points 1 + 2 + 3 + 4. Please see the general comments.
>
> 5. Thanks for pointing out the work of Heinze-Deml et al. (2017). This paper assumes the algorithm is provided with datasets of different environments, each one has a fixed value of E. In our paper, we focus on a vanilla SCM, where the algorithm is only provided with observational samples of X and Y = g(X,E) (i.i.d samples). The samples are not divided into subsets that are invariant w.r.t E.
>
> In addition, the two independence tests are different. In our case, we require that E is independent of X, while in papers, such as, (Heinze-Deml et al. (2017); Zhang et al., 2011) they assume that Y is independent of E given X. This assumption generally fails in our setting. We will note it to the related work section in the next version of the paper.
>
> We made a considerable effort to extend GPI, LiNGAM, Direct-LiNGAM, and CDS to the multivariate case, however, these algorithms and/or their existing implementations are highly dependent on the assumption that the data is univariate. Fortunately, we were successful in extending PNL and added the results to the table.
>
> We also added empirical results on the MOUS-MEG real-world dataset.
>
> 6. Regarding CGNN, we found a bug in the public implementation of it that disabled the training to run on the GPU. It is now fixed and we have added the results of running CGNN to the table.

---

> > ### Comment · AnonReviewer4 · 2019-11-14
> > **Authors' Additions Warrant Score Increase**
> >
> > It looks like pretty much all of my questions about the univariate case were answered in the general comment. The below quote from the authors seems important:
> >
> > "In order to avoid raising unnecessary antagonism, we mentioned our criticism in a soft manner. We are aware that this point was probably missed by the reviewers and we will make a significant effort to emphasize it more in the next version."
> >
> > Indeed, I completely missed this. It seems, then, that one of your main contributions should be that you point out that evalutating these methods in the univariate case is bad because of how trivial it is (comparing marginals is enough). If this is a main claim, then I think the paper could benefit from having more emphasis on this. I see that the authors have uploaded an edited version that does exactly this. Because the authors have such an extensive experimental analysis (many different datasets) in the univariate case, I find this claim, which important for those doing research in this area, sufficiently substantiated by the experiments in this paper.
> >
> > Because of the above, the addition of an evaluation of PNL and CGNN in the multivariate setting, and the addition of MOUS-MEG real-world dataset dataset for the multivariate setting, I am updating my score to a 6.

---

### Author Response · Authors · 2019-11-13
**General comments for all of the reviewers**

We would like to thank the reviewers for your constructive feedback, we appreciate it.  We revised the paper according to the reviews and uploaded it.

We would like to provide a few general comments to questions on the univariate case that were raised by the reviewers.

In the first part of the paper, we provide a critical stand-point to the univariate SCM. We claim that the univariate SCM is too simplistic. To do so, we show empirically that one is able to infer the causal relationship between two random variables X -> Y without checking the relationship between them. The intention to do so is explicitly stated in the abstract, introduction and summary:

Abstract: “We show that by comparing the individual complexities of univariate cause and effect in the Structural Causal Model, one can identify the cause and the effect, without considering their interaction at all.”

Intro: “In this work, we demonstrate that for the 1D case, which is the dominant case in the existing literature, the SCM model leads to an effect that has lower complexity than the cause. Therefore, one can identify the cause and the effect by measuring their individual complexities, with no need to make the inference based on both variables simultaneously. Thus, the decision as to which of the two is the cause and which is the effect may not be based on causality but on complexity.”

Summary: “...its success in predicting cause and effect indicates an inherent bias in the unidimensional datasets.”

Almost all of the algorithms in the literature try to compare the success of mapping from X to Y (and vice versa) under various conditions (independence tests, complexity, etc’). We introduce the heuristic AEQ method and show empirically that one can infer the causal relationship between X and Y, simply by comparing their complexities, without comparing any mappings between them. The complexity of a random variable X is measured by the MSE error produced by an autoencoder that tries to map T(X) to itself. Here, T(X) is the transformation of X into a multivariate random variable (see Sec. 3.1, page 3).

It is important for us to emphasize that we are not advocates of the AEQ method. It is given as an indication that the univariate SCM framework is too simplistic to capture the true notion of causality and that a method that obviously does not check causality between random variables is able to get competitive results on several benchmarks.

In order to avoid raising unnecessary antagonism, we mentioned our criticism in a soft manner. We are aware that this point was probably missed by the reviewers and we will make a significant effort to emphasize it more in the next version.

Next, we wanted to ground the AEQ algorithm in a theoretical manner. The intention is to be able to claim that the univariate model is inherently too simplistic. To do so, first, we showed that the reconstruction error of an autoencoder trained on samples of a multivariate r.v U is proportional to the entropy of U (see Lem. 1).

Then, we informally said that the entropy of Y is supposed to be smaller than that of X. This claim holds in the discrete case, where Y = f(X) (with no noise involved). We agree with the reviewers that this claim is generally false (when noise is involved and the r.v.s are continuous). Note, that in our case, we do not compare between the reconstruction errors of X and Y, while we compare them for T(X) and T(Y). Therefore, the AEQ does not compare the entropies of X and Y, it compares the entropies of T(X) and T(Y). For a discrete r.v X, we still have h(T(X)) >= h(T(Y)) for Y = f(X), where f is a monotonic function.

Since it is unclear to us if this inequality hold in the general case, we decided to take down the discussion regarding the data-processing inequality. However, we do think that Lem. 1 is important since it provides a better understanding of what is measured by the AEQ. We are very thankful for the reviewers for pointing out these issues.

Finally, we compare the AEQ method to a comparator of the standard Shannon entropy of X and Y and show that the entropy does not indicate the causal direction. Note that this does not contradict the combination of Lem. 1 and the empirical results of the AEQ. That is because the AEQ compares the MSEs of autoencoders trained on T(X) and T(Y). By Lem. 1 the reconstruction errors are proportional to h(T(X)) and h(T(Y)) and not to h(X) and h(Y).  It is also worth mentioning that when running our experiments we tried different alternatives to the above T. For other transformations we achieved much worse results. Therefore, we believe that the success of the combination between the quantiles and the autoencoder is not accidental.

Multivariate case:
We have added multiple empirical results for the multivariate case, following the reviews. In addition, we added a new real-world dataset we call MOUS-MEG, which is described in the experiments section.

---

### Decision · Program_Chairs · 2019-12-19

**Decision:**

Reject

**Comment:**

The author response and revisions to the manuscript motivated two reviewers to increase their scores to weak accept. While these revisions increased the quality of the work, the overall assessment is just shy of the threshold for inclusion.